# Threats in Water–Energy–Food–Land Nexus by the 2022 Military and Economic Conflict

G.-Fivos Sargentis [1,*], Nikos D. Lagaros [2], Giuseppe Leonardo Cascella [3] and Demetris Koutsoyiannis [1]

1 Laboratory of Hydrology and Water Resources Development, School of Civil Engineering, National Technical University of Athens, Heroon Polytechneiou 9, 15780 Athens, Greece
2 Institute of Structural Analysis and Antiseismic Research, School of Civil Engineering, National Technical University of Athens, 15780 Athens, Greece
3 Department of Electrical Engineering and Information Technology, Politecnico di Bari, 70126 Bari, Italy
* Correspondence: fivos@itia.ntua.gr

**Abstract:** The formation of societies is based on the dynamics of spatial clustering, which optimizes economies of scale in the management of the water–energy–food (WEF) nexus. Energy and food are determinant measures of prosperity. Using the WEF nexus as an indicator, we evaluate the social impacts of the current (2022) conflict and in particular the economic sanctions on Russia. As Russia and Ukraine are major global suppliers of energy sources, food, and fertilizers, new threats arise by their limitations and the rally of prices. By analyzing related data, we show the dramatic effects on society, and we note that cities, which depend on a wider area for energy and food supplies, are extremely vulnerable. This problem was substantially worsened due to the large-scale urbanization in recent decades, which increased the distance from food sources. We conjecture that the Western elites' decision to sanction Russia dramatically transformed the global WEF equilibrium, which could probably lead to the collapse of social cohesion.

**Keywords:** water–energy–food nexus; land; war; economy; infrastructures

## 1. Introduction

### 1.1. The Role of Water–Energy–Food–Land Nexus and Clustering to Prosperity

There are different meanings of prosperity. Some think that it is measured in money, others relate it to pleasure and life satisfaction, while others link it to spirituality. However, it could be argued that the basic human needs related to water, energy and food (WEF) compose a nexus that is not only necessary for the survival of humans, but is able to explain their prosperity as well [1]. This nexus is extended by the addition of land, as land is a fundamental source for the support of WEF.

Decision making in the modern world is driven by economic aspects and monetarist policies that ignore the basic needs in the WEF nexus. However, water, energy and food are not derived by money; rather, money and economic growth derive from the availability and the access to WEF. In other words, even though modern thinking believes that money is the basis of the economy, the availability of the WEF nexus is the necessary condition for the existence of society (and economy), not the reverse.

In prehistory, humans discovered new energy sources and created water infrastructures to support the transition from hunter–gatherers to farmers. This gave them the ability to cluster in smaller spaces such as villages and, later, cities. The increase in clustering was a stride of civilization [2,3], but cities always depended on external resources (e.g., in antiquity, Athens imported wheat from the area of the Black Sea [4,5]).

As the WEF nexus is critical for human survival, and the abundance of resources is connected to life expectancy [6], we have to assume that an optimization of the WEF nexus management is required, and it is crucial to minimize the resources for production. In this aspect, we have to examine the parts of the nexus and their interplay.

Currently, humanity is facing a major challenge: the rapidly growing demand for WEF. Population growth, the different ways of life of each society, and the urgent need to improve WEF security for the poorest are putting increasing pressure on resources [7]. Unless there are significant changes in production and consumption patterns, if we would like to support the rate of growth and current way of living, agricultural production should increase by about 60% until 2050, and global electricity production is projected to increase by about 60% over the next ten years [8,9]. Thus, a careful management of the WEF nexus is required. This fact has emerged in the literature over the last 10 years (Figures 1 and 2) [10–19].

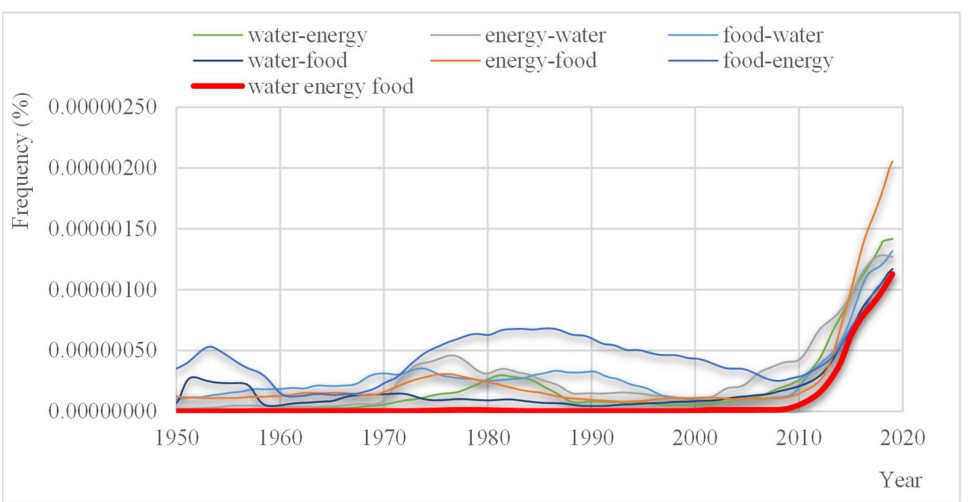

**Figure 1.** Frequency of appearances of the indicated phrase in Google Books up to year 2020 [20]. Data adapted graphically by Google ngrams (data from [21]).

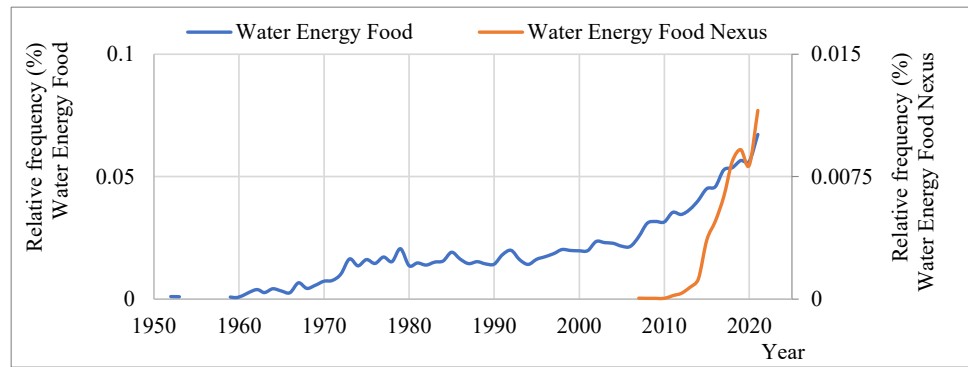

**Figure 2.** Relative frequency of appearances of the indicated key phrases in the article titles, abstracts and keywords of about 70 million articles written in English, which are contained in the Scopus database up to 2021 (data from [22]).

Notably, the WEF nexus management is already part of the evolution process of homo sapiens. Related papers have overviewed the relationship, through the evolution process, of living organisms with the following determinant items:

- walking on two feet was an energy-saving step [23]
- the function of the human brain is more energy-efficient than in animals [24].

## 1.2. The Elements of War

Carl Von Clausewitz, a Prussian general and theorist of war, in his classic book *On War*, defines war as: "continuation of politics carried on by other means." [25].

Dalio, in his book *Principles for Dealing with the Changing World Order* [26], describes several parameters that are responsible for the thriving of civilization such as education,

technology, and more. Merging them describes the dynamics that formulate the pursuit of humans for wealth and "The Big Balance of Power Cycle That Drives the Big Peace/War Cycle".

Many studies assume that the cause of war could be ideological aspects (e.g., religion) [27,28] or the control of natural resources [29,30], but history has shown many wars for ridiculous reasons [31]. Mark [32] notes:

> *The tribe mentality always results in a dichotomy of an 'us' vs. a 'them' and engenders a latent fear of the 'other' whose culture is at odds with, or at least different from, one's own. This fear, coupled with a desire to expand, or protect, necessary resources, often results in war.*

A general target of war is the domination and the conquest of an enemy's land [33]. As the WEF nexus depends on land uses, war makes the WEF nexus vulnerable, and its final management will be affected by the war's outcome. The WEF nexus has also been used as weapon in the widely known scorched-earth policy [34], which has broad social and environmental impacts [35]. Furthermore, since antiquity, castle sieges were based on the limitation of water and food [36,37].

In 2022, Russia's ongoing special military operation (SPO) in Ukraine and the Western reaction in the form of economic sanctions has created limitations in terms of access to energy and food for the West.

The cities of the Western world could be considered the modern castles. As energy and food supplies are based on a global equilibrium, we see correlations with the archetypical siege method of antiquity, as the diachronic target is the WEF nexus. We note that water infrastructures have not been impacted by the SPO, but the limitations of the two other parts of the nexus (energy and food) have created similar issues of restrictions.

In this paper, we attempt to understand what is actually happening with the limitations of the WEF nexus by Russia's SPO.

## 2. Methodology

In general, we consider that WEF availability is a fundamental issue for the prosperity of society. Therefore, in this paper, using the WEF and land nexus as an indicator, we attempt to clarify how the limitations of the nexus, which were caused by Western sanctions on Russia, affect society in the following steps.

- The correlation of energy consumption with gross domestic product (GDP) per capita and life expectancy is our first indicator. The bulk of the extant literature has assumed a causal relationship between energy consumption and GDP and life expectancy [38–46]. By another viewpoint, economics dominated by the narrative of ecology [47] question the role between energy and prosperity [48–53]. However, in our approach, using recently available data (2022), we find correlations that yield clear trends. Using them as indicators, we estimate the effect of energy consumption changes in GDP per capita and life expectancy.
- The archetypical measure of wealth is not gold, silver or money. As wheat is the base element of the digestive menu of humans, an acceptable method to evaluate real wealth is by the estimation of the wheat wages (i.e., the liters of wheat that can be bought by a daily wage) [6,54–62]. This allows us to create an important comparison of economic wellbeing. The prices of wheat in countries' markets differ from the prices of global markets; however, by assuming that local retail prices are double the global wholesale ones, we find an indicative relation of wheat wages based on the average GDP per capita. With this method we evaluate the effect on real wages according to the alterations of wheat prices.
- In an attempt to assess whether current Western policies incorporate rationality, by using publicly available data, we evaluate the limitations and the strategies that are presented as a defense of the availability of the WEF nexus (fertilizers and novel installation of renewable-energy infrastructures).

- Clustering is a method of growth and is supported by large-scale infrastructures. Partitioning is a method of protection and can be applied to many different threats such as viruses (social distancing), wars, and wildfires [2,63]. Using satellite images and publicly available data, we evaluate by Climacogram Integral the evolution of spatial clustering in Europe (1990–2010) using Hurst–Kolmogorov dynamics [2,64–71]. Urbanization is reflected in the increasing trend of cities' clustering, which we have observe lately. This leads to a transformation of social structures, guiding people far from the WEF nexus, making societies vulnerable to WEF nexus limitations.

The summary of the methodology is depicted in Figure 3. A symbolic representation of the prosperity of humankind is depicted in center of Figure 3. The WEF–Land nexus is depicted in the left image of Figure 3. The dynamic of cities' clustering is depicted by the night-light satellite image (right image of Figure 3).

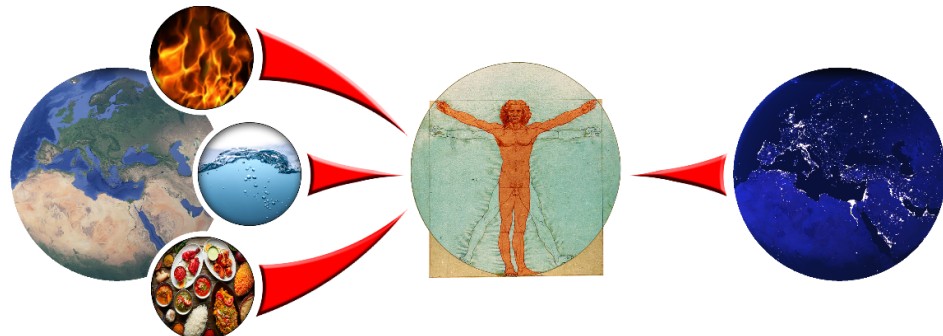

**Figure 3.** An attempt at an artistic representation of issues of prosperity. WEF–Land nexus (**left**) and the dynamics of cities' clustering by night-light satellite image (**right**).

In summary, we assert that at present, these facts have not been perceptible by Western elites. Even though huge datasets are currently publicly available from the International Monetary Fund, World Bank, Our World in Data, Energy Information Administration, Maddison Project Database, Gapminder, National Oceanic and Atmospheric Administration, Water Footprint network and Food and Agriculture Organization of the United Nations, decision making is customarily based on "imaginary beliefs" [72]. In our study we highlight some critical "imaginary beliefs" about the WEF nexus and we evaluate them based on publicly available data. The data sources used are cited in the related parts of the paper.

## 3. Russia's Special Military Operation in Ukraine 2022

### 3.1. The Cause of Rurssia's SPO

Russia has argued that its SPO in Ukraine has been caused by the decision of the Ukrainian government to join the Western defensive alliance NATO, as well as Russia's decision to protect people subjected to, what they called, eight years of the Ukrainian government's bullying and hostility towards Russian people in east Ukraine [73,74].

Western countries, the US and the majority of European Union (EU) countries have imposed economic sanctions on Russia with the hope that they will punish the Russian Federation. In this vein, the West has stopped buying Russian resources assuming that this will stop the finance of the SPO. The efficiency of economic sanctions is always questionable. Economic sanctions were implemented by Austrians on Serbia (1906–1909), closing the border to Serbian pork (widely known as the "pig war"). This economic war was a counterproductive measure as Serbia quickly found other export markets [75].

In Western thought, money is superior in value to the WEF nexus, and its elements are considered tradable, which definitely obey the laws of the market. That is why more and more efforts are being made to translate them into stock products, with disastrous consequences for social cohesion and the economy, as happened with the policies of dictator Pinochet in Chile for the privatization of water [76].

### 3.2. The Demolition of Supply Chain

The four major grain traders (Archer Daniels Midland, Bunge, Cargill, Louis Dreyfus), which control 90% of the global grain trade [77], refused to exit Russia despite mounting pressure due to the economic sanctions [78]. However, until now (August 2022) there has been "no smooth sailing for grain via the Black Sea" [79].

The break in the supply chain has caused the prices to increase. Figure 4 shows a more detailed match between wheat and oil prices in January 2010–March 2022. An IMF report depicts how the SPO in Ukraine is reverberating across the world's regions [80]. A relevant speech of Josep Borrell (officer of the EU for Foreign Affairs and Security Policy/Vice-President of the European Commission), describes how "Russia's invasion of Ukraine puts the global economy at risk" [81].

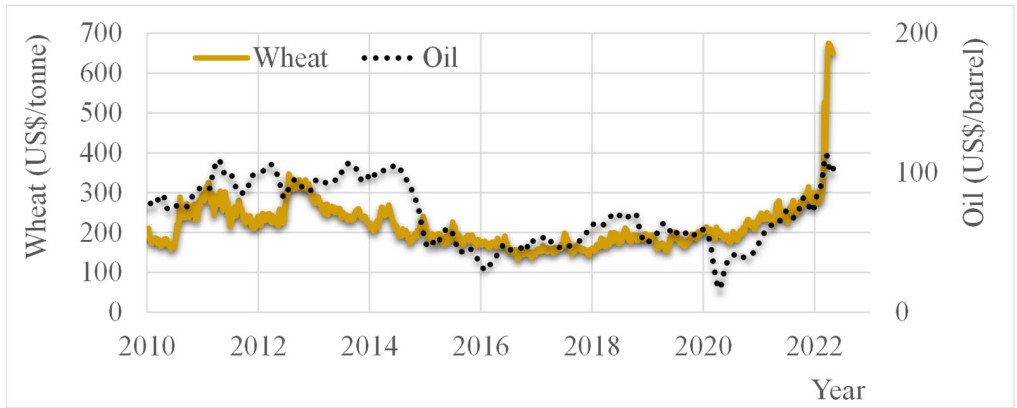

**Figure 4.** Prices of wheat (data from [82]) and oil (data from [83]) (January 2010, June 2022).

Sun Tzu in *The Art of War* noted [84]: "Make forays in fertile country in order to supply your army with food". The description of the interactions in Ukraine is difficult when we consider that the flat-land anaglyph and the economies of scale made Ukraine one of the largest granaries of the world, connecting it with global equilibrium. Based on a relevant report of the UN [85], the Secretary-General of United Nations António Guterres urges world leaders to "Act now to end food, energy and finance crisis" [86].

## 4. Evaluating the Limitation of Energy and the Rally of Food Prices

It is clear that the SPO has transformed the land, changing the equilibrium of energy resources [87]. Energy is connected to the production of all materials and goods. The energy needed for the creation of a material or product is referred to as embodied energy [88–91]. Energy is also connected to the production of food [92–102]. For example, intensive crops that have a high yield of food require 25,000 MJ/ha per year (indicative value that depends on the type and method of cultivation). In addition, water is also connected to energy prices, as many irrigation systems are high energy consumers that use pumps or desalination.

### 4.1. Energy

Figure 5 shows that in many countries in Africa, the energy surplus is lower than human energy, which means that the living standards are similar to prehistoric ones. Therefore, we have to consider that money does not determine the cost of the kWh, but it is the access to the kWh that determines the value of money. Thus, the dramatic increase in energy cost in the first half of 2022 in Europe [103] could be considered one of the reasons for the devaluation of the euro in this period [104]. At the same time, according to the American network CBS NEWS, Russians, by connecting their energy sources to their currency, made it "the strongest currency in the world this year" [105].

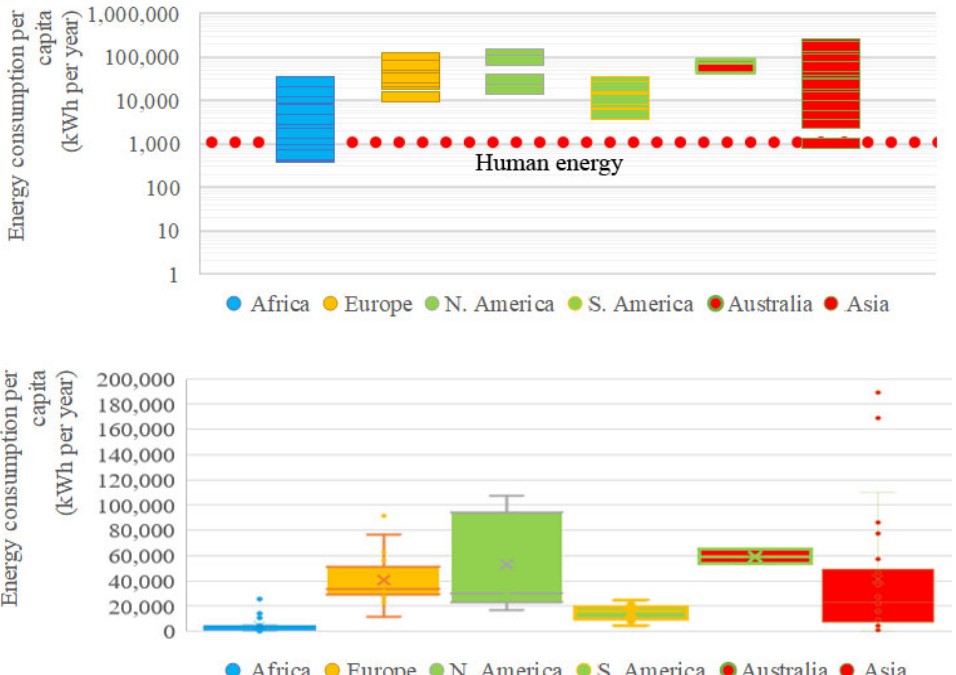

**Figure 5.** Annual energy consumption per capita in different countries (data from [106]). Top: plot in logarithmic scale indicating the human energy; Bottom: box plot.

Even if the causal link of energy consumption to GDP and life expectancy is questionable according to some researchers, by correlating recent global data on the current energy consumption with GDP per capita and life expectancy, we identify interesting trends (Figure 6a,b).

According the regression equations shown in Figure 6, if Europe reduces its energy consumption [107] (for example by 20%), this would mean that the population would lose about one year of life expectancy and about 15% of income. However, if the reductions are not equally distributed (for example, if the lower classes bear the cost of the energy poverty and restrict their consumption to 50%), this means that they would lose about 3 years of life expectancy and about 40% of income.

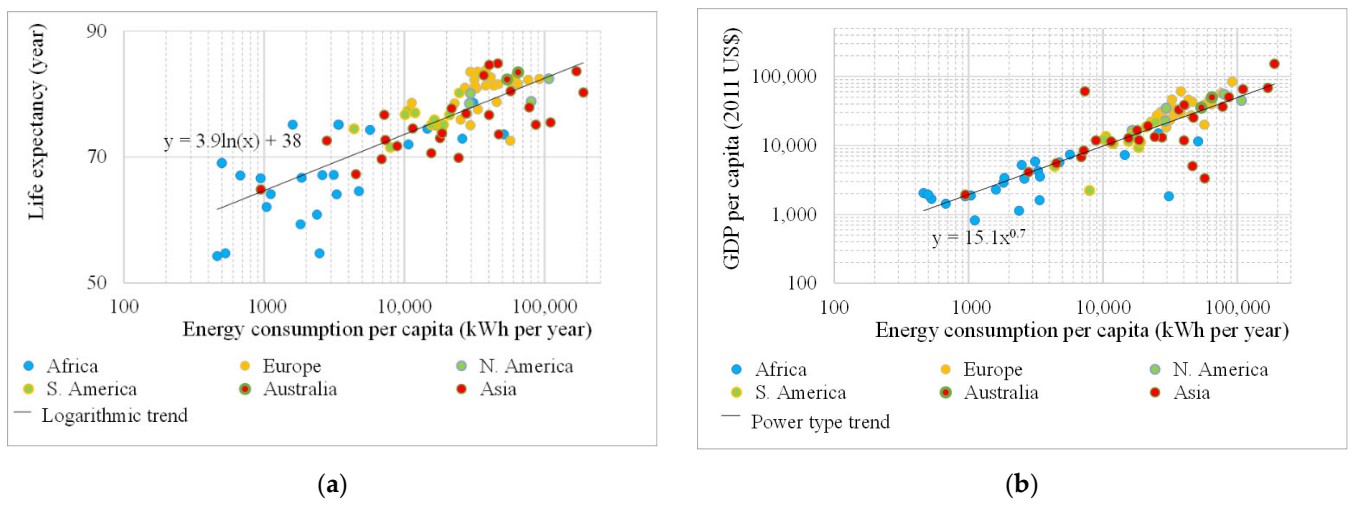

**(a)**

**(b)**

**Figure 6.** Energy consumption (2020) related to: (**a**) life expectancy; (**b**) GDP per capita (data from [106,108]).

### 4.2. Food

Russia, along with Ukraine, produces 15 percent of the world's wheat [109] (others think more [110]), supplying various dependent countries. Wheat is the base element of the digestive menu of humans; therefore, by correlating the daily wage with liters of wheat (wheat wages), we have a stable measure for estimating wealth.

In 2015, the limit of extreme poverty was a daily wage of 1.9 US$ [111], which corresponded to about 6.5 L of wheat, somewhat lower than the average wage in antiquity, which was about 8 L of wheat). It is estimated that the average wage in antiquity was close to what we use to characterize extreme poverty today [112,113]. In 2022, the limit of extreme poverty was considered a daily wage of 3.2 US$ [114], but as the values were changed after February 2022, this corresponds to about 4 L of daily wheat wage.

In order to estimate how the Ukrainian crisis has affected the world, we translate the average daily wage per country and globally into wheat using the Maddison Project Database 2020 [108], which provides data on the average GDP per capita in 169 countries up to 2018. We use the average GDP per capita in 2016 with the average wheat price in 2010–2020 (about 210 € per ton) and the average GDP per capita in 2018 as a benchmark for 2022, with the wheat price of June 2022 (about 650 € per ton).

With this adaptation (Figure 7), looking at the data by region, it appears that in 2022, regions such as Africa and part of Asia corresponded to an average living standard that is lower than the pre-industrial era (which was considered the limit of extreme poverty). By adopting wheat wages in income mountains, and adopting the GDP per capita in 2021 as a benchmark for 2022 (Figure 8), we see that the average living conditions of the of pre-industrial societies in 2016 did not exceed 17.5%, while in 2022 this percentage rocketed to 61%. It should be noted that the UN, in the aforementioned report of 8 June 2022 [85], estimates that 25% of the world's population is threatened with famine. In a related article, *The Economist* describes this as "The coming food catastrophe" [115].

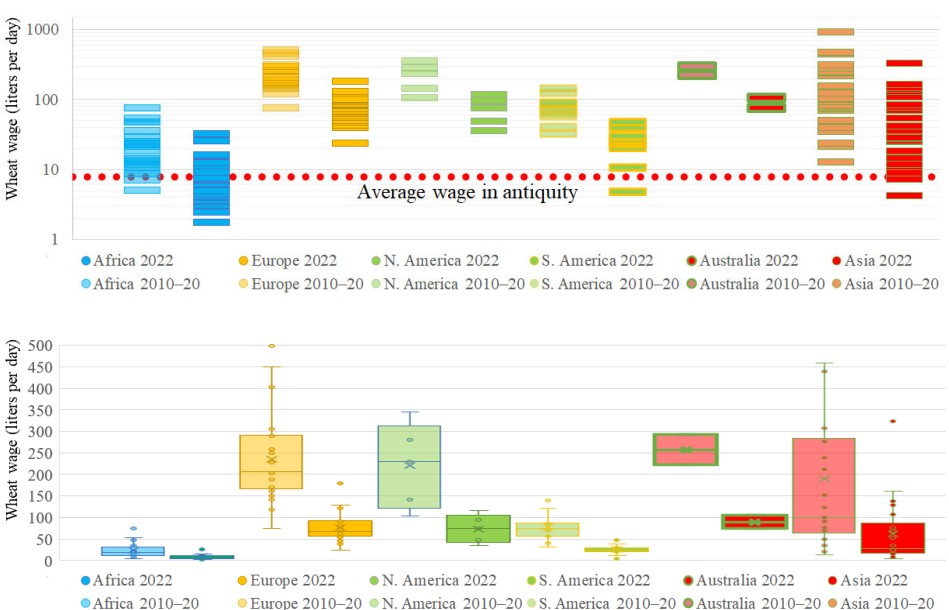

**Figure 7.** Average wages in different countries (data from [108]) converted to wheat wages. 2016: wheat price average of 2010–2020 was about 210 € per ton; 2022: wheat price was about 650 € per ton. Data accuracy is low as prices depend on local markets. Top: plot in logarithmic scale indicating the average wage in antiquity (e.g., what we used to consider as extreme poverty); bottom: box plot.

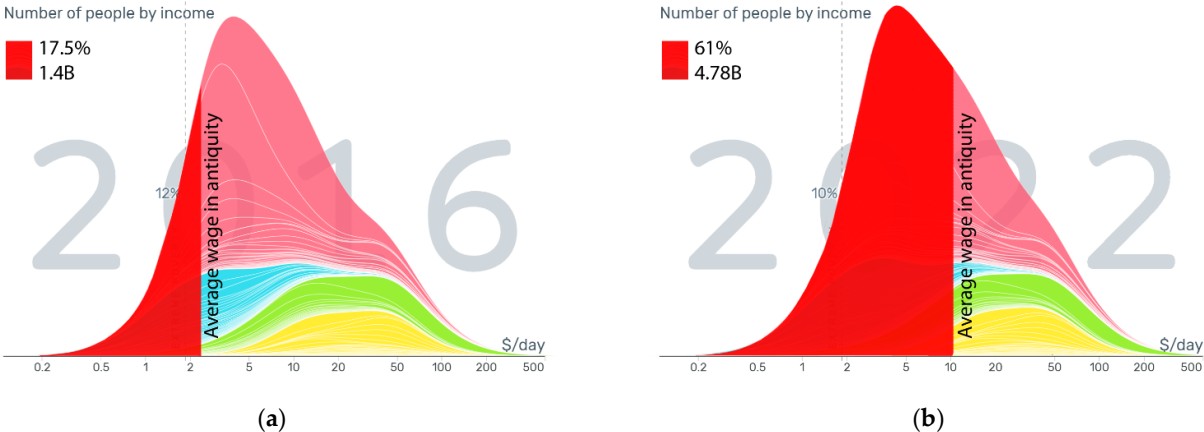

**Figure 8.** Income mountains (data adapted from [116] after transformation) marking the average wheat wage in antiquity. (**a**) 2016: wheat price average of 2010–2020 was about 210 € per ton; (**b**) 2022: wheat price was about 650 € per ton. Data accuracy is low as prices depend on local markets.

Figure 7 shows that, adopting GDP per capita of Europe and North America in 2022 as wheat wage is similar to the GDP per capita of South America (2010–2020) before the Ukrainian crisis.

## 5. The Acceleration of "Green Growth" as a Defensive Method of West

The collective Western thought is dominated by a Malthusian [117] perspective. The dominant narrative obeys the limits of growth and the alarm of overpopulation [118,119], degrowth [120–122], and "Green experiments" [123,124], even if related reports have highlighted that these lead to a costly failure [125,126].

Jordan Peterson [127] notes:

*Richer people care about "the environment"–which is, after all, outside the primary and fundamental concern of those desperate for their next meal.*

Social unrests caused by "Green" goals such as fertilizer and energy limitations, which lead to a rally of energy–food cost, have already emerged in the Netherlands [128], Sierra Leone [129], Peru [130], Argentina [131], Sri Lanka [132] and other countries [133].

The conflicts of the present have led to two great goals of "Green Growth": the limitation of fertilizers and the replacement of the lost energy from Russian natural gas and coal, which was caused by Western economic sanctions on Russia, with renewable-energy infrastructures.

### 5.1. Fertilizers

A major "Green" goal is the limitation of fertilizers. Russia is the largest supplier of fertilizers. The most recent update given by Our World In Data (2014) [134] shows that Russia is responsible for 25% of the world's production of fertilizers, while it consumes about 10% of its production [135].

It is estimated that in 2015, 3.54 billion people (about half of the global population) were fed because of the productive surplus value generated by fertilizers [136–138].

No one can say with certainty what percentage of agricultural production will be lost because of Russia's restricted fertilizer exports. However, by correlating the above, we can estimate that Russian fertilizers are used to produce food for about 1 billion people (1/8 of the global population).

Fertilizers are high-energy-consuming products (1.2% of the world's total energy on an annual basis) [139]. Russia uses its energy surplus for the production of fertilizers, which highlights the links between energy and food.

*5.2. Energy*

The EU President Ursula von der Leyen stated on 18 May 2022 that the EU will remedy energy poverty with another "Green" goal; renewable-energy installations [140]. However, even if the EU could create these infrastructures, it has to find their embodied energy (the energy to create them). In orders of magnitude, the embodied energy needed for a wind turbine per MW is 30 TJ, while the energy produced for a 20-year period is about 700 TJ or 35 TJ annually [141]. The embodied energy needed per MW for a solar panel, is 7 TJ while the energy produced for a 20-year period is about 112 TJ or about 6 TJ annually [142] (only for the creation of the equipment, not the transportation and installation).

The consumption of primary energy by the EU in 2020 was 15,482 TWh. The share of primary energy from wind and solar energy sources in Europe for 2020 was estimated to be about 17.95%. The primary energy from wind is 395 TWh and comes from the installed capacity of 217 GW (wind turbines), and solar is 146 TWh, which comes from the installed capacity of 168 GW (solar panels) [143].

In 2020, the EU consumed 3799 TWh of natural gas (29–40%; about 1350 TWh from Russia [144,145]). Therefore, it has yet to build more than 2.4 times the installed capacity of the renewable-energy installations that had been built in the last 20 years. If we consider, for example, that the EU should expedite an investment in the installation of an energy mix similar to the existing mix, then the energy investment will be more than 18 million TJ, which means that with the current high energy prices in the EU (may increase further) will cost tens of trillion €.

Ioannidis and Koutsoyiannis [146] note: "it is evident that, in the long term, RE projects will indeed be the cause of massive landscape changes. It is the first time in human history that energy generation has so high land-use demands". The estimation of the land use for wind energy is 33,3333 m$^2$/MW [147] and for solar energy is 31,970 m$^2$/MW [148], which means that 191,000 km$^2$ are needed for the assumed energy mix.

## 6. Clustering Conflicts of Land Use

Clustering is a natural process that allows organisms to use natural resources more efficiently. Figure 9 shows that elephants require about 10,000 times less energy per mass than a mouse. Larger scales also increase the efficiency for mammals in terms of water consumption for survival.

A more holistic inspection of natural evolution reveals additional elements of clustering. Dinosaurs were the largest living creatures in nature, but about 66 million years ago they disappeared. Smaller animals such as mammals survived because of "being small If you're small you probably have a large population and thus a wider genetic diversity" [149]. In this sense, even clustering can be important for the economy and is associated with goals of growth and partitioning as a means of protection, by reducing the dependencies and risks associated with a centralized infrastructure [2].

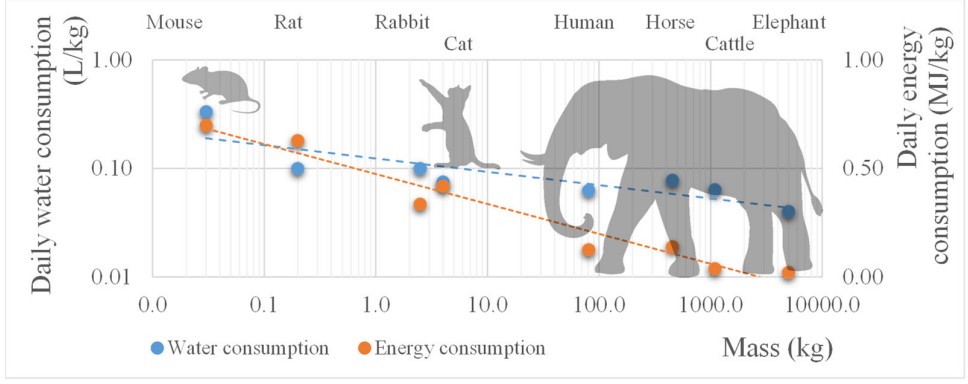

**Figure 9.** Daily energy and water consumption of mammals (data from [150]).

In the last 30 years, the model of the growth of EU countries has focused on urbanization (Figures 10 and 11a), which is strongly connected to secondary and tertiary economic sector, and not directly linked to the WEF nexus [151]. However, cities are individual clusters that highly depend on the WEF nexus from wider areas. Specifically, 75% of global primary energy [152] and 80% of global food is consumed in urban areas [153], while the urban population is only 56% [154].

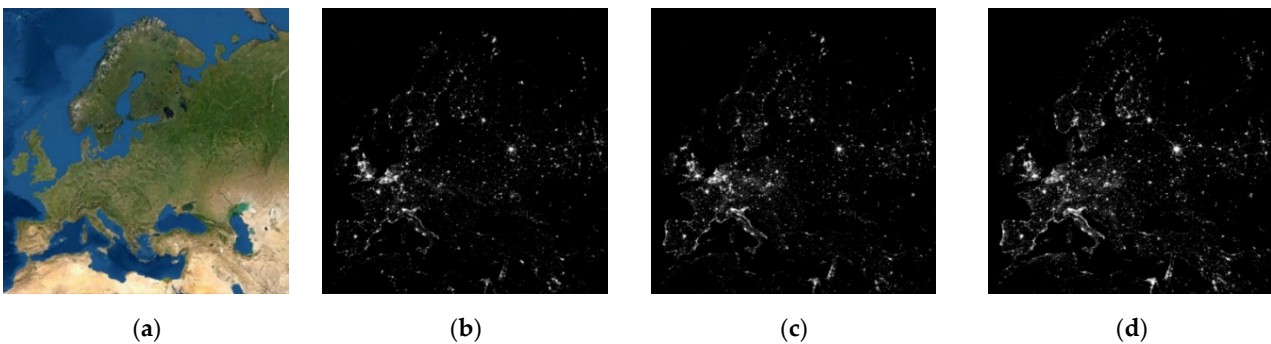

(**a**)  (**b**)  (**c**)  (**d**)

**Figure 10.** (**a**) Europe in Mercator projection. Night lights of Europe in Mercator projection: (**b**) 1992; (**c**) 2002; (**d**) 2012 [2,155].

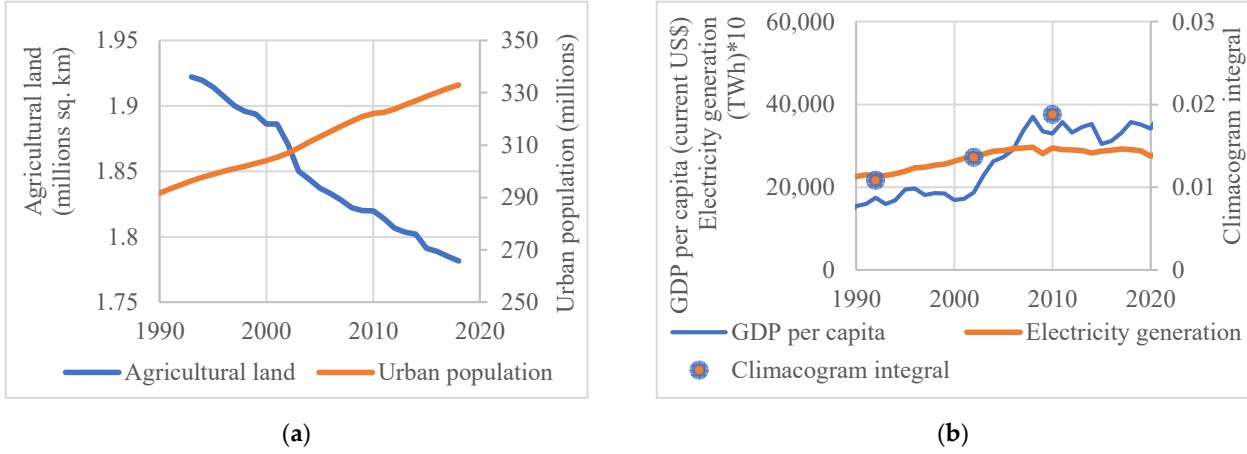

(**a**)  (**b**)

**Figure 11.** EU (1990–2020) (**a**) Agricultural land (data from [156]) and urban population (data from [157]); (**b**) GDP per capita (data from [158]), electricity generation (data from [106]), and clustering indicator (Climacogram integral).

By inspecting the trend of Europe's land uses (Figure 11b), we note the increasing in cities' clustering, which is depicted in Figure 11b by Climacogram Integral [2]. In the same diagram, we see the evolution of the GDP per capita and the electricity generation.

## 7. Discussion

In the introduction, we discussed the necessity of the growth of energy and food production in order to stabilize our current way of living [8,9]. However, the evolution of wheat and energy prices shows an opposite trend.

In order to investigate the potential for survival in the current crisis, we can focus on the unexpected and unfortunate real-world experiment: the COVID-19 lockdown in 2020, which caused an unprecedented decrease in carbon emissions that characterize energy consumption [159], leading energy prices to fall.

Lockdowns can indeed reduce energy consumption; however, they cannot solve the problem of food, as during the COVID-19 lockdown, the supply chains of food were functional. Therefore, in the present conditions, societies have to build self-sufficient

social structures, probably by reversing urbanization, which could bring people closer to food sources.

In this case, the social structure could be shaped by energy and food collectives in order to optimize the scale of energy and food production. Xanthopoulos [160–162] coined the term "endogenous productive reconstruction" to describe sustainable wealth, which could lead to self-sufficiency and potentialities of growth. Describing the social transformation Stathis Stavropoulos [163] notes: "*τα όνειρα δεν είναι η ποίηση της Ουτοπίας, αλλά η ποίηση της Ανάγκης*" (dreams [of social reconstruction] are not the delivered by Utopia, but by Necessity).

However, the reversal of urbanization needs water resources [164]. The limitation of water resources could make life very difficult for the new inhabitants [165,166] who (especially in the West) have learned to live with an abundance of water (Figure 12).

A related report of the United Nations on the human right to water and sanitation notes that [167]:

> *The water supply and sanitation facility for each person must be continuous and sufficient for personal and domestic uses. These uses ordinarily include drinking, personal sanitation, washing of clothes, food preparation and personal and household hygiene. According to the World Health Organization (WHO), between 50 and 100 litres of water per person per day are needed to ensure that most basic needs are met and few health concerns arise.*

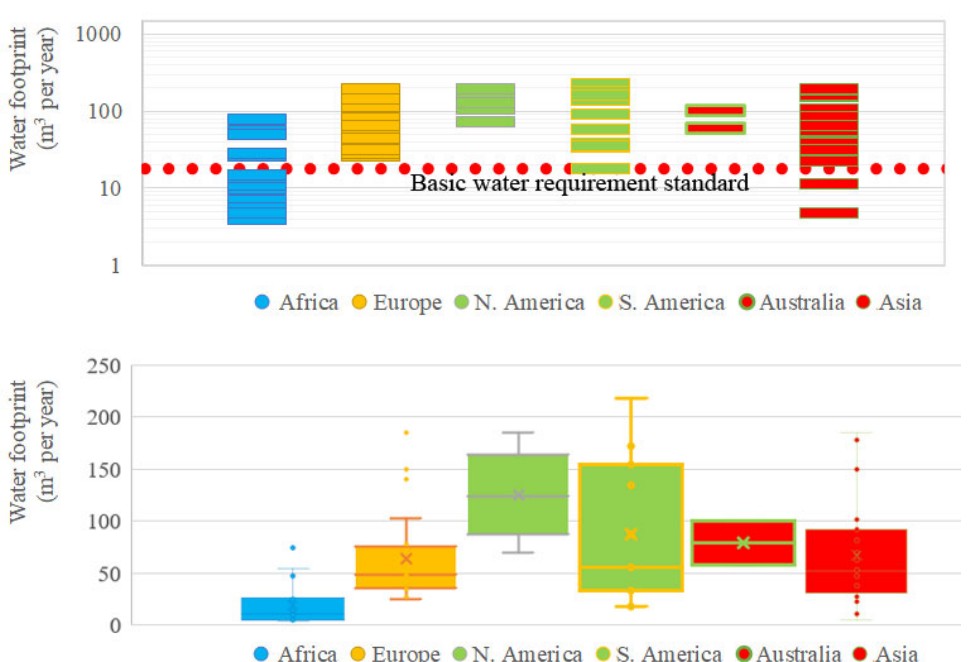

**Figure 12.** Annual water footprint (consumption of domestic blue water) per capita in different countries (2005) (data from [168,169]). Top: plot in logarithmic scale indicating the basic water requirement standard; bottom: box plot.

## 8. Conclusions

The structure of modern societies is based on the dynamics of clustering (i.e., large cities) and on the creation of large-scale infrastructures for the support of the WEF nexus, as economies of scale make this more efficient. Sargentis et al. [2] note:

> *While human social clustering increases the chances for social progress and prosperity, it also increases exposure and vulnerability to different kinds of risk. For the first time in human history more people live in cities than in rural areas. This rapid growth in the number of people living in cities and urban landscapes is increasing the world's susceptibility to natural disasters and other threats.*

One characteristic fact that shows the vulnerability of cities has been highlighted in this period with the Russian SPO, which has caused limitations to the access of the WEF nexus due to the economic sanctions from the West on Russia. Even though geopolitical aspects show that the availability and prices of resources differ among countries, the presented data on the variation of energy consumption and wheat prices show that (in orders of magnitude) a recession phase seems inevitable. In addition, interactions between the nexus elements resulting in an alteration of the equilibrium of the nexus could be the trigger for a social and economic collapse [170–173], as the sociopolitical complexity of the modern world blurs the larger perspective [174]. Obviously, societies cannot manage complexity if they are in spiritual decadence [175], which is probably the reason why Sir Simon Jenkins [176] described Western sanctions against Russia as:

> . . . *the most ill-conceived and counterproductive policy in recent international history.*

Likewise, Escobar [177] notes:

> *The Collective West Self-Justification Show staged to obliterate its ritualized suicide offers no hint of transcending sacrifice implied in a ceremonial seppuku. All they do is to wallow in the adamant refusal to admit they could be seriously mistaken.*

Energy-saving lockdowns and working from distance may again be imposed. However, as the supply chains have been broken, there is a need for a restructuring of the clustering of cities in a manner that guarantees self-sufficiency.

In this effort, the West has to expedite a social and land transformation, which could be achieved in areas with available food and water resources where people could move from cities to the countryside, perhaps making additional work remotely where possible.

The summer of 2022 has been salted with questionable news. It seems like Western elites [178] are feeding the Ukrainian battlefield with new weapons, considering that they will win these battles by military force, even if they have already lost the battle of the WEF nexus. It appears completely absurd that the Western elites insist to new energy sanctions which will obviously lead Western societies to more energy poverty [179]. As the elites live on the rightmost edge of the income mountain's curve (Figure 8), they likely believe that they will have enough money to overpass this crisis, ignoring the fact that (as the WEF nexus is the real wealth) money could lose its value, and they cannot buy something that does not exist. Interestingly, two major representatives of the Western elites of the past, Kissinger [180] and Roberts [181], have serious objections about the way things are going.

The result of the induced limitations on the WEF nexus is sobering and may become tragic. It appears that if the Western elites forge a society that will not cover people's real needs (WEF), then the collapse of the social cohesion of the West is a strong possibility.

**Author Contributions:** Conceptualization G.-F.S.; methodology G.-F.S.; validation G.-F.S. and D.K.; formal analysis, G.-F.S.; investigation, G.-F.S.; data curation, G.-F.S.; writing—original draft preparation, G.-F.S., N.D.L., G.L.C. and D.K.; writing—review and editing G.-F.S., N.D.L., G.L.C. and D.K.; visualization, G.-F.S.; supervision, D.K.; project administration, N.D.L. All authors have read and agreed to the published version of the manuscript.

**Funding:** This research was funded by European Union funds, grant number 101007595.

**Data Availability Statement:** The databases that have been used are referred to in detail in the citation given in the text and are publicly available.

**Acknowledgments:** This research was supported by the ADDOPTML project: "ADDitively Manufactured OPTimized Structures by means of Machine Learning" (No: 101007595) belonging to the Marie Skłodowska-Curie Actions (MSCA) Research and Innovation Staff Exchange (RISE) H2020-MSCA-RISE-2020.

**Conflicts of Interest:** The authors declare no conflict of interest.

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
