# Peer review of "Threats in Water–Energy–Food–Land Nexus by the 2022 Military and Economic Conflict"

_land, doi:10.3390/land11091569_

Round 1
Reviewer 1 Report
This study discusses “Limitations in Water-Energy-Food and the vulnerability of the cities. Case study: Ukraine 2022”. It has certain theoretical significance and research value for understanding the Water-Energy-Food nexus. The organizational structure of the paper is good. However, the following questions need to be solved by the authors.
1. The Water-Energy-Food nexus is a worldwide scientific problem,but I didn't see the literature of more scholars in this paper.
2. As we know, there are four major grain traders (Archer Daniels Midland, Bunge, Cargill, Louis Dreyfus) affecting 80% of the global grain trade. Is this inconsistent with the parts of "The demolition of supply chain" ?
3. The resolution of some figures (like Figure 9) needs to be improved.
4. Water-Energy-Food nexus is a closely related coupled mutual feed system. The development of one subsystem often needs to consume the resources of the other two subsystems. Therefore, what implications will the conclusions of this paper provide for the collaborative optimization management scheme for stakeholders ?
Reviewer 2 Report
I recommend reconsidering the title of the article. The case study connected to Ukraine 2022 create a marginal not substantial part of the article.
Due to the growing popularity of the topic, I would expect a more extensive discussion.
By my opinion the subsection "... The cause of Rurssia’s SPO" containing political opinions does not belong to the scientific journals of this focus.
Reviewer 3 Report
The authors intended to discuss the vulnerability of cities in terms of their Water-Energy-Food consumptions using the example of Ukraine in 2022. Although the topic would be interesting, the information presented in the article seemed to be scattered and did not contribute to addressing the same questions. The conclusions were not sufficiently or logically supported.
Reviewer 4 Report
Journal: Land
Manuscript ID: land-1859687
Type: Article
Title: Limitations in Water-Energy-Food and the vulnerability of the cities. Case study: Ukraine 2022
In this current report, authors try to illustrate the impact of Ukraine war as a case study on WEF. The topic is very interesting and very important. However, authors failed in presenting their work in an appropriate way! I have the following crucial point:
1- Writing style: this work cannot be considered as a research article; it is just an article for a newspaper
2- Research goal is not clear
3- Research hypothesis never mentioned.
4- Introduction, method, and result are not clear.
5- Title is not informative
6- Abstract is just a simple story about WEF, there is no clear goal, method, or result!
7- Clear statistical analysis is messing.
If authors want to improve their work, I can suggest the following:
1- The introduction section should be rewritten with special emphasis on the WEF and war.
2- Clear hypothesis and goal should be added
3- Research gape should be mentioned
4- Method section should be added with special focus on: data collecting, analysis, and statistical approach.
5- Results should be more concrete.
It is important to sperate each section, adding figures in every place of the article is confusing. Not mentioning the data sources is misleading. All in all, this article is not meet the minimum criteria to be published.
Round 2
Reviewer 4 Report
In this version, authors did some changes. However, the method and writing style did not meet the criteria to be publish as a scientific research article. Some examples:
"In this paper, we are trying to understand what is actually happening with the limitations of WEF nexus by Russia’s SPO” it is not a goal of a research article !
“However, in our approach, using recently available data, we find correlations which gives us clear 111
trends. Using them as indicators, we estimate the effect of energy consumption changes in GDP and life expectancy”
Using recently available data: from where? and how the data was used? What is the variable ? time span?
Using them as indicators? Indicator for what?
we estimate the effect of energy consumption changes in GDP and life expectancy: how? What method?
L120: “we find an indicative relation of wheat wag…” how ?
L123 “In an attempt to assess whether current Western policies incorporate rationality evaluate by publicly available data” what data, and how this part is conducted!
Fig 3. Is not sound
“The result of the induced limitations on WEF nexus is sobering and may become tragic. We are not optimistic that any change will be made with this paper and we have the fear that possibly the recovery will come after an absolute collapse. But we feel it as our scientific duty to highlight these issues.” We don’t include our hopes in the method section!
Round 3
Reviewer 4 Report
Dear authors, I dont see any changes in the method sections. My raised issue remains as it is.